# The *Escherichia coli* Amino Acid Uptake Protein CycA: Regulation of Its Synthesis and Practical Application in l-Isoleucine Production

**DOI:** 10.3390/microorganisms10030647

**Published:** 2022-03-17

**Authors:** Christine Hook, Natalya Eremina, Petr Zaytsev, Daria Varlamova, Nataliya Stoynova

**Affiliations:** Ajinomoto-Genetika Research Institute, 117545 Moscow, Russia; natalya_eremina@agri.ru (N.E.); petrzaytsevrf@gmail.com (P.Z.); daria_varlamova@agri.ru (D.V.); nataliya_stoynova@agri.ru (N.S.)

**Keywords:** CycA, transport, transcription regulation, leucine-responsive regulatory protein (LRP), cyclic AMP receptor protein (CRP), microbial producers, amino acid uptake, l-isoleucine, metabolic engineering

## Abstract

Amino acid transport systems perform important physiological functions; their role should certainly be considered in microbial production of amino acids. Typically, in the context of metabolic engineering, efforts are focused on the search for and application of specific amino acid efflux pumps. However, in addition, importers can also be used to improve the industrial process as a whole. In this study, the protein CycA, which is known for uptake of nonpolar amino acids, was characterized from the viewpoint of regulating its expression and range of substrates. We prepared a *cycA*-overexpressing strain and found that it exhibited high sensitivity to branched-chain amino acids and their structural analogues, with relatively increased consumption of these amino acids, suggesting that they are imported by CycA. The expression of *cycA* was found to be dependent on the extracellular concentrations of substrate amino acids. The role of some transcription factors in *cycA* expression, including of Lrp and Crp, was studied using a reporter gene construct. Evidence for the direct binding of Crp to the *cycA* regulatory region was obtained using a gel-retardation assay. The enhanced import of named amino acids due to *cycA* overexpression in the l-isoleucine-producing strain resulted in a significant reduction in the generation of undesirable impurities. This work demonstrates the importance of uptake systems with respect to their application in metabolic engineering.

## 1. Introduction

Amino acids (AA) are attractive and promising biochemicals, with applications ranging from use as animal feed additives to food flavors and pharmacy nutrients. Currently, *Escherichia coli* along with *Corynebacterium glutamicum* and *Pantoea ananatis* is widely used as a host bacterium for the industrial production of amino acids [1,2,3,4,5,6,7,8]. Metabolic engineering approaches based on rational design are commonly used, including the deregulation of biosynthetic genes, enhancement of amino acid efflux, and modification of central metabolism, aimed at the elevation of building-block synthesis [9,10,11]. Despite the fact that *E. coli* is the most well-characterized microorganism, the study of new features of its physiology and genetics is of particular interest from the viewpoint of metabolic engineering.

Membrane transport regulation is an important step in the development of producer strains [10]. Membrane transport refers to the collection of mechanisms providing the passage of different substances and energy across a cell membrane, a process that is essential for cell viability. Nearly 20–30% of all genes within a genome are functionally involved in transport. Amino acids are the basic units of proteins and also appear as intermediates in regulatory processes; thus, they play a crucial role in cell metabolism. Systems for amino acid import are widely found in bacteria. Instead of endogenously engaging in the power-consuming synthesis of amino acids, the cell can import them for subsequent utilization in protein synthesis or catabolism processes [12]. The transport of amino acids is an active transport process against a concentration gradient, which is mediated by specialized proteins in prokaryotic cells. These carriers are divided into two groups: primary and secondary transporters. The proteins from the first group are represented by the superfamily of ATP-binding cassette (ABC) transporters, which use the energy from ATP to move components across a membrane against the gradient. Such carriers are efficient in the case of a low substrate concentration in the environment. So-called secondary transporters use the electrochemical gradient as a source of energy and usually work with rather high concentrations of the transported compound. Such proteins are classified as members of the amino acid/polyamine/organocation (APC) superfamily, which is one of the largest groups and comprises about 250 transporters [13]. Many amino acids are transported through APC carriers that function as solute–cation symporters (e.g., the lysine importer LysP) or antiporters (e.g., the methionine exporter YjeH) and solute–solute antiporters, such as the glutamate/GABA importer GadC [14,15,16].

While amino acid uptake is obviously beneficial for a growing cell, when exploiting microbes for the production of amino acids, their reuptake is undesirable, resulting in a futile cycle of production and uptake from the culture broth. This may result in a waste of metabolic energy and decreased production rates. In contrast, the physiological significance of amino acid excretion in microorganisms is less obvious, but more attractive for metabolic engineering of strains for hyperproduction of a particular amino acid. Generally, in metabolic engineering, efforts for transport optimization are focused on the search for and application of new specific amino acid exporters, because the efficiency of the export is important for final accumulation of target compound [12,17,18,19]. In some cases, enhancement of export is absolutely necessary for successful producer breeding due to toxicity of the amino acid of interest, as it is for cysteine, for example [20]. However, importers may also be used for the improvement of amino acid production, considering the necessity to control the accumulation of undesirable amino acid impurities in the culture broth. In the last 20 years, more than 10 amino acid import systems have been identified in *E. coli*, including the aromatic amino acid uptake system AroP [21], isoleucine importer BrnQ [22], aspartate/glutamate carrier GltIJKL [23], and high-affinity uptake system specific for phenylalanine [24].

The search for new transporters and the study of their substrate specificity are usually carried out using various approaches, such as the analysis of amino acid transport kinetics [25], analysis of amino acid transport inhibition by some substances [26], and selection of mutants resistant to some toxic substrates with subsequent analysis of the involved mutations [27]. The toxic analogues of amino acids might be used for this purpose; therefore, transporters for such substances are highly likely to be able to transport the amino acid corresponding to the analogue [28,29].

Several decades ago, Cosloy et al. revealed that a mutation in the *cycA* locus abolished the toxic effect of d-serine on cell growth [28] and assumed that the gene *cycA* (b4208) was responsible for d-cycloserine transport in *E. coli*, as was earlier shown by Wargel et al. [29]. Further investigations of the amino acid transport systems expanded the range of the putative substrates of CycA transporter, which was found to include glycine (Gly), d-alanine (d-Ala), and β-alanine (β-Ala) [26,30].

CycA is a 470 aa transmembrane protein localized in the inner cell membrane. Computer analysis based on phylogenetic results and on the sequence alignment of APC superfamily proteins placed CycA in the amino acid transporter (AAT) family [13]. According to the Transporter Classification Database (TCDB), CycA belongs to the APC superfamily class of electrochemical potential-driven transporters [13,31]. The experiments of Schneider et al. demonstrated that β-Ala transport was dependent on an electrochemical H^+^ potential, which could be disrupted by CCCP (carbonyl cyanide *m*-chlorophenyl hydrazone) addition; accordingly, CycA was proposed as the major β-alanine transporter [30].

The regulation of *cycA* gene expression is poorly understood. The discovered mechanism relates to the regulation of CycA mRNA translation. According to the literature, small regulatory RNA (sRNA) GcvB and protein factor Hfq affect *cycA* translation [32]. There is minimal data relevant in elucidating its transcription regulation. High-throughput analysis methods allowed proposing the *cycA* gene as a member of regulons of transcription regulators such as Lrp (leucine-responsive regulator protein), Crp (cAMP receptor protein), ArcA (anoxic redox control), and Nac (nitrogen assimilation control) [33,34,35,36,37]. Lrp and Crp are global regulatory factors that are known to be involved in the control of numerous cell processes, including catabolism and the transport of small molecules such as sugars and amino acids (as reviewed in [11]). Typically, high-throughput analysis data require confirmation by means of a more careful analysis of the observed effects and interactions using different reporter systems and other approaches.

In this study, we found that, in addition to amino acids already identified as CycA substrates, several aliphatic amino acids (e.g., l-valine (Val), norvaline (Nva), α-aminobutyrate (AABA)) can be imported by this permease. Next, we analyzed the influence of transport of different substrates on the expression of a reporter gene controlled by the native regulatory region of the *cycA* gene. The influence of some transcription factors, including Lrp and Crp, on the *cycA* gene expression level was studied. Evidence for the direct binding of Crp to the *cycA* regulatory region was obtained. With practical relevance to this study, the modulation of the uptake of known and newly identified CycA substrates through the oversynthesis of this permease was shown to be a feasible strategy for simplifying the whole process of the microbial production of l-isoleucine (Ile).

## 2. Materials and Methods

### 2.1. Strains, Plasmids, and Media

All bacterial strains and plasmids used in this study are listed in Table 1. The following media were used to culture bacteria: lysogeny broth (LB), M9, super optimal broth (SOB), and Super Optimal broth with Catabolite repression (SOC) [38]. Glucose (0.4%) was added to minimal medium as a carbon source. The antibiotics ampicillin (Ap, 100 mg/L), chloramphenicol (Cm, 20 mg/L), and kanamycin (Km, 50 mg/L) were used when necessary.

### 2.2. DNA Manipulation

Genetic manipulation of *E. coli* and techniques for the isolation and manipulation of nucleic acids were performed according to standard protocols [38]. Taq polymerase and 1 kb DNA Ladder were purchased from Thermo Scientific Inc. (Waltham, MA, USA). Oligonucleotides were purchased from Evrogen (Moscow, Russia). The sequences of the oligonucleotide primers are presented in Appendix A.

### 2.3. Construction of Strains

Insertions and deletions in the chromosome of *E. coli*, typical of the MG1655 K12 strain, were prepared via λ-Red modification according to the method of Datsenko and Wanner [43]. The plasmid pKD46, carrying the arabinose-inducible λ-Red genes and kindly gifted by Dr. Wanner, was used. Combining necessary cassettes within one strain was conducted using the P1 transduction method [46]. The details of mutant strains construction are presented in the Appendix A.

### 2.4. Growth Curves

The growth of *E. coli* strains in M9 glucose medium with different supplements (one of the sets of amino acids or keto acid) was studied using a biophotorecorder (model TVS062CA; Advantec Toyo Roshi Kaisha, Ltd., Tokyo, Japan). Inhibitory concentrations of amino acids and keto acids were selected by sensitivity assay performed on agar plates based on literature data [17,30,47]. The strains MG1655 and M1 (MG1655 *cat*-P_L_-*cycA*) were grown in LB medium overnight. The cells were harvested, washed twice in buffered saline, and resuspended in the appropriate medium. The cells were cultured in parallel in L-shaped glass tubes at 37 °C; the initial OD_660_ was 0.15 at the zero timepoint, and OD_660_ values were measured every 10 min in the growing culture for 40 h.

### 2.5. Amino Acid Uptake Assay

A modified amino acid uptake assay was performed according to previously reported methods [14]. Strains MG1655 and M1 (MG1655 *cat*-P_L_-*cycA*) were grown in LB medium overnight. The cells were harvested, washed twice in buffered saline, and resuspended in N^−^ minimal medium (modified M9 minimal medium without nitrogen source) supplemented with one of the sets of amino acids, at 37 °C, to an optical density at 600 nm (OD_600_) of 0.2, and then the extracellular amino acid concentration was measured using an Agilent 6000 at the timepoints 1, 2, 4, and 6 h. The uptake rate (mM/mg DW) was calculated as the consumed AA (the measured extracellular AA concentration was subtracted from the initial one) normalized to the dry weight (DW, mg) value. To calculate DW, the k = 0.54 factor measured for *E. coli* strain MG1655 grown in minimal M9 medium was used.

### 2.6. β-Galactosidase Activity Assay

Cells were grown to the mid-logarithmic phase in LB or M9 medium. The M9 medium was additionally supplemented with the required amino acid (10 mM). The activity of β-galactosidase was measured according to Miller’s method [48]. The mean of triplicate experiments is presented.

### 2.7. Electromobility Shift Assay (EMSA)

EMSA was performed using a previously reported method with some modifications [49]. Fluorescently labeled DNA probes were amplified using 5′-TAMRA (λ_ex_/λ_em_ = 544/576 nm) labeled and standard primers (P8–P13). Briefly, a crude extract of *E. coli* cells (200 µg) or pure Crp protein (12 ng) was incubated with 50 ng of TAMRA-labeled DNA probes in a binding buffer (10 mM Tris–HCl, pH 7.5, 1 mM EDTA, 100 mM KCl, 1 mM DTT, 10 g/mL BSA, 5% glycerol, and 20 µM cAMP). The mixtures were incubated at room temperature for 30 min, and the protein-bound DNA substrate was separated from free DNA probe on a 6% native polyacrylamide gel using 1× TAE (40 mM Tris acetate, pH 8.0, 2 mM EDTA) by electrophoresis at 4 °C. Electrophoresis was performed at 80 V for 3 h, and the gel was imaged using a Typhoon 9410 (GE Amersham Molecular Dynamics).

### 2.8. CRP Extraction and Purification

Crp was purified from *E. coli* SA500/pHA5 cells (kindly provided by Prof. Mironov A.S.) using the Ni-NTA chromatography method. This strain possesses a plasmid pHA5 with a *crp* gene that enhances Crp synthesis [50]. Several His residues present in the protein sequence provide affinity binding to Ni^2+^; hence, the addition of a His-tag is not necessary [51]. Protein elution was performed in several rounds using a series of imidazole solutions of consistently increasing concentration (30–150 mM). At first, unspecific bound proteins were eluted; Crp was mainly present in the late fraction. Samples from these fractions were analyzed by SDS polyacrylamide gel electrophoresis (according to the standard Laemmli method under denaturing conditions). Western blot analysis was performed as follows: the protein was electrotransferred to a Hybond-C membrane using a Trans-Blot Semi-Dry Transfer Cell (Bio-Rad, Hercules, CA, USA) according to the manufacturer’s recommendations and probed with rabbit peroxidase-coupled polyclonal anti-CRP antibody (MyBioSource, San Diego, CA, USA). The blots were visualized with an ECL Prime Western Blotting System (GE Healthcare, Chicago, IL, USA).

### 2.9. Test-Tube Cultivation Conditions

For each of the tested conditions of Ile accumulation, the strains were grown in LB medium at 37 °C overnight; then, 0.1 mL of each culture was inoculated into 2 mL of fermentation medium with the required supplements (20 µM chlorsulfuron or 4 g/L of threonine) in a test tube, which was then cultivated for 72 or 96 h at 32 °C on a rotary shaker (250 rpm) until all of the glucose was consumed. Next, the amino acid concentration in culture broth was measured using a CAMAG TLC Scanner 3. Results are the average of four independent assays. The composition of the fermentation medium was as follows: glucose 60 g/L, (NH_4_)_2_SO_4_ 15 g/L, KH_2_ PO_4_ 1.5 g/L, MgSO_4_ 1 g/L, thiamin 0.1 g/L, LB 1/10 (*v*/*v*), and CaCO_3_ 25 g/L; the pH was adjusted to 7.0.

## 3. Results

### 3.1. Novel Substrates of the CycA Transporter Include Branched-Chain Amino Acids

In *E. coli*, the *cycA* gene encodes a membrane protein belonging to the APC superfamily [31]. CycA is known to import glycine, d-serine (d-Ser), d-cycloserine, and l-, d-, and β-alanine. We examined whether CycA functioned as an importer of some other amino acids that are similar in structure to l-alanine (Ala), e.g., Val, l-leucine (Leu), Ile, and AABA (see Figure 1).

The evaluation of the influence of the set of amino acids was performed by assessing the cell growth rate in minimal medium. We performed a growth test using different amino acids, Val, AABA, Ile, and azaleucine (Azaleu, analogue of Leu), as well as a precursor of Ile, α-ketobutyrate (2KB), since they were able to decrease the cell growth rate in minimal medium [17]. The mechanism of such an effect is explained by the inhibition of some key biosynthetic enzymes and the subsequent interruption of biosynthesis of essential compounds or the incorporation of structural analogues of amino acids into proteins that contributes to the formation of partially active or inactive enzymes and leads to growth inhibition or even lethality of microorganisms [52]. The initial wild-type MG1655 and *cycA*-overexpressed strain M1 (MG1655 *cat*-P_L_-*cycA*; all strains used in this study are presented in Table 1) were cultured in minimal medium (M9) or in the same media containing the inhibiting concentrations 50 mg/L β-Ala, 15 g/L Ile, 50 mg/L AABA, 500 mg/L azaleucine, 0.5 mg/L Val, or 150 mg/L 2KB. The growth curves are presented in Figure 2. The strains grew similarly in M9. All the compounds negatively affected the cell growth of the strain MG1655; however, this effect was more pronounced in the case of strain M1. Since β-Ala is a known substrate of CycA [30], this experiment served as evidence of CycA oversynthesis by strain M1. Poor growth of this strain in the presence of β-Ala in comparison with the growth curve of MG1655 indicates the strengthened import of β-Ala and, thus, the efficiency of the construct *cat*-P_L_-*cycA* present in the M1 strain. The results shown in Figure 2 demonstrate that *cycA* overexpression improved the sensitivity of the strain to the toxicity of the explored compounds. These data suggest that CycA can also import such substrates.

For further investigation of the CycA substrate specificity, the amino acid uptake was assayed. The cells were initially incubated in medium with 50 mM Ala (or other CycA substrates), and the extracellular AA level was measured. To calculate the uptake rate, the amount of consumed AA (the measured concentration was subtracted from the initial level) was normalized to the culture OD. As shown in Figure 3, the uptake rate was higher in the case of the M1 strain, especially for Ala and AABA. This effect was less pronounced for other branched-chain amino acids (BCAAs, e.g., Val, Ile, Leu).

On the basis of the results of these two experiments, it was concluded that BCAAs and 2KB were imported by CycA.

### 3.2. CycA Gene Expression Is Induced by Addition of Its Substrate(s) into Culture Medium

It was reported that the expression level of some transporter-encoding genes depends on the concentration of their corresponding substrates [17]. To study the expression of *cycA*, we constructed the strain M2 (MG1655 *cat*-P*_cycA_*-*lacZ*) possessing the translational fusion of the *cycA* regulatory region containing the first 33 nt of the coding sequence (from −223 to +117 relative to the TSS (transcription start site)) containing sequences upstream and downstream of the TSS (P*_cycA_*) with the *lacZ* reporter gene. With the individual addition of extracellular amino acids (10 mM), the effects of the substrate amino acids on reporter gene *lacZ* expression were investigated by measuring β-galactosidase activity. We tested well-known CycA substrates (Ala, Gly, d-Ser) as well as the newly identified ones (AABA, Val, Nva, Ile, Leu, 2KB). In one particular case, for evaluation of the influence of Val addition, we used a specific strain M3 constructed on the basis of a Val-resistant strain K12 2Δ P_L_-*ilvBN*^fbr^ by introducing the cassette *cat*-P*_cycA_*-*lacZ*, because 10 mM Val addition suppressed the cell growth in minimal medium and prevented the estimation of the influence of Val addition on the reporter gene expression.

As can be seen, the expression level changed with the addition of CycA substrates (Figure 4). As expected, the activity was downregulated (1.3-fold) in the presence of Gly. As known, *cycA* is an object for negative regulation by sRNA GcvB [32]; therefore, the increased Gly level led to *gcvB* overexpression and, hence, to a decreased level of reporter translation. A range of tested aliphatic AAs were able to upregulate the expression level of the P*_cycA_*-controlled reporter. The most significant effect was demonstrated by the addition of Leu (twofold increase). Similarly, elevated P*_cycA_* transcription levels were observed in the presence of l-alanine and l-isoleucine. This is consistent with the literature indicating the *cycA* gene as a member of the Lrp regulon family [33] and the ability of different aliphatic amino acids—not only Leu—to bind to Lrp and modulate its activity [53]. This result indicates that *cycA* expression was induced by the addition of Leu, Ile, Ala, Nva, AABA, and d-Ser, and repressed by the addition of Gly and 2KB.

### 3.3. Prediction and Verification of Potential cycA Gene Expression Regulators

We analyzed the regulatory region P*_cycA_* using the DNA–protein interaction database DPInteract [54] and bioinformatic software “Sequence Editor and Tools” [55]. According to the DPInteract, several putative transcription regulators were listed (ArcA, Crp, FarR, Fis, GlpR, HNS, IHF, Lrp, OmpR, RpoD, RpoS, SoxS, and TyrR). Several putative regulator-binding sites were confirmed by the “Sequence Editor and Tools” software. For further investigation, the following factors were chosen: RpoS, Crp, Lrp, IHF, HNS, and FarR. A schematic view of the binding sites of these transcriptional regulators is presented in Figure 5.

For the examination of the effects of the predicted transcription factors on *cycA* gene expression, we obtained a set of strains with the individual deletions of these transcription factor genes in combination with the cassette *cat*-P*_cycA_*-*lacZ*. In particular, the strains M4–M10 (see Table 1 for genotype) were obtained. The effects of a lack of transcription factors on reporter gene *lacZ* expression were investigated by measuring β-galactosidase activity in cells at the exponential stage of growth. We used the strain M2 (MG1655 *cat*-P_cycA_-*lacZ*) as a reference. As the regulatory effect of GcvB on *cycA* expression is well known, a similar experiment as that used to establish this [32] was carried out with *cat*-P_cycA_-*lacZ*, in which the M4 (∆*gcvB*) strain was used to verify the reliability of our results.

In line with the in silico predictions, deletion of some regulatory genes affected the β-galactosidase activity of the *cat*-P*_cycA_*-*lacZ*-harboring strains (Figure 6). Thus, most of the predicted factors appeared as expression activators (RpoS, Crp, IHF, HNS), except for FarR, which did not influence the rate of P*_cycA_* expression. Given that these transcription factors are global regulators, the effect of gene deletions may be indirect. Therefore, additional investigations of the regulation mode are necessary.

Another predicted dual transcription factor is LRP (leucine-responsive regulator protein). As the LRP action is dependent upon its binding with Leu, we decided to measure β-galactosidase activity in the strain M10 (∆*lrp*) grown in minimal medium with or without Leu addition (Figure 7). As it can be seen, Lrp negatively regulates *cycA* in the absence of leucine. Binding of Lrp with Leu increased expression in the *lrp*+ (M2) strain, whereas the deletion of *lrp* resulted in expression becoming independent of Leu. These data are consistent with the literature [33], where *cycA* was proposed to belong to an Lrp regulon based on the results of microarray analysis. The model proposed by the authors of that paper suggests that Lrp represses *cycA* transcription at the exponential growth phase when grown in minimal medium, and the addition of leucine addition abolishes the repression effect.

### 3.4. Crp Role in Regulation of cycA Gene Expression

One of the transcription factors predicted to regulate *cycA* gene expression is CRP (cAMP receptor protein). Crp is known to be a global regulator responsible not only for catabolism of substrates under carbon starvation conditions, but also the transport of small molecules that could be utilized as a carbon source, including some amino acids [34]. The results of the primary analysis defined Crp as an activator of *cycA* expression in the exponential growth phase. The microarray analysis did not provide us with evidence of a direct influence of Crp on the *cycA* gene transcription [34]. Thus, one cannot exclude that Crp may be involved in a whole cascade of interactions with the participation of other regulatory factors affecting the expression of this gene.

Firstly, there is a possibility that the *cycA* gene expression is modulated by some other factors whose activity is controlled by the Crp level. In particular, Crp is known as a repressor of Hfq protein synthesis. Hfq is a major factor necessary for the interaction of small regulatory RNA (sRNA) with the target mRNA [32,56]. As mentioned, the *cycA* gene expression is subject to negative regulation by sRNA GcvB. This allows Crp to affect Hfq-mediated processes as an activator [57]. The following scenario can be suggested: repression of Hfq synthesis by Crp removes repression of *cycA* translation mediated by GcvB in complex with Hfq, thus leading to the activation of CycA synthesis.

To clarify the role of CRP in CycA synthesis regulation, we constructed strains harboring a transcriptional fusion of the P*_cycA_* regulatory region with the *lacZ* reporter gene (ΔP*_lac_*::*cat*^exc^-P*_cycA_*-5′-UTR*_lacZ_*-*lacZ*). This construct contains the 5′-UTR of the *lacZ* gene that lacks binding sites for GcvB sRNA and is, thus, independent of GcvB- and Hfq-mediated regulation. We performed the β-galactosidase assay using strains M11 and M12 with transcriptional fusion (ΔP*_lac_*::*cat*^exc^-P*_cycA_*-5′-UTR*_lacZ_*-*lacZ*) in a *crp*^wt^ or Δ*crp* genetic background, respectively (Figure 8). The obtained results demonstrate that CRP still acts as an activator, and its participation in *cycA* gene transcription regulation can be supposed. However, only evidence of a direct interaction of Crp with the promoter region can confirm Crp as a transcriptional regulator of *cycA* gene expression.

### 3.5. Binding of Protein Factors with the cycA Regulatory Region

In order to study the direct binding of transcription factors to the *cycA* regulatory region, we performed the electrophoretic mobility shift assay (EMSA). The assay was conducted with a fluorescently labeled fragment of P*_cycA_* and a crude extract of *E. coli* cells. The 208 bp length probe was designed to contain predicted binding sites for Crp, Lrp, and RpoS factors (from −271 to −63 relative to the TSS). It was obtained by the amplification of the *E. coli* K12 MG1655 chromosome fragment using TAMRA-labeled primers. After the binding reaction with the crude extract of *E. coli* K12 MG1655 cells, the reaction mix was applied to the PAAG. Visualization of the TAMRA-labeled probes alone or in complex with proteins in gel was achieved using fluorescent scanner Typhoon 9410 (GE Amersham Molecular Dynamics). The obtained results revealed a set of proteins, including Crp, Lrp, RpoS, and others, that might bind to P*_cycA_* (Figure 9). This is consistent with our suggestion of the existence of some transcriptional regulators of the *cycA* gene.

In parallel, we performed a control experiment using a fluorescently labeled fragment of the *lacZ* gene promoter region (P*_lacZ_*). The 218 bp length (from −238 to −20 relative to the TSS) contains Crp and LacI binding sites [58]. To prevent LacI binding, 1 mM IPTG was added to the reaction mix. In this sample, we observed a protein–DNA probe complex (Figure 9), which we expect to contain Crp. A band with similar electrophoresis mobility was present in the lane with the P*_cycA_* probe. Moreover, gel retardation of the P*_cycA_* binding reaction using a crude extract from the isogenic strain, but Δ*crp* demonstrated a lack of the corresponding band. This may indicate that Crp directly binds to the regulatory region of the *cycA* gene.

Considering that the Crp molecular weight is 46 kDa, we posited that lower bands may correspond to complexes with RpoS (37 kDa) and Lrp (19 kDa). In order to clearly identify the proteins, EMSA with solutions of pure transcription factors needed to be performed.

### 3.6. Crp Binds cycA Gene Regulatory Region

To confirm the direct influence of Crp on *cycA* gene transcription, we performed EMSA experiments using purified Crp and *cycA* promoter region fragments.

The pure protein was applied to EMSA with DNA oligonucleotides corresponding to different fragments of the *cycA* regulatory region (Figure 10). Fragment 1 (208 bp length, from −271 to −63 relative to the TSS) included sites for several transcription factors. To precisely identify the binding region specific for CRP, we also used a shorter fragment 2 (73 bp length, from −136 to −63 relative to the TSS), whereas, as a negative control, we used *cycA* regulatory region fragment 3 (148 bp length, from −271 to −123 relative to the TSS) without a predicted CRP-binding motif. A scheme of the regulatory region and its fragmentation is shown in Figure 10. Gel retardation of the DNA probes was observed in samples with pure Crp and fragment 1 or fragment 2. The probe with fragment 3 (without a predicted Crp-binding site) did not show changes in electrophoretic mobility upon Crp addition. In conclusion, we confirmed the presence of a CRP-binding site in the predicted area.

### 3.7. Overexpression of cycA Decreases Accumulation of Impurities by an l-Ile-Producing Strain

The modification of transport systems usually affects the accumulation of different compounds in the culture broth. In the case of importers, such an impact may concern the level of impurities. To evaluate the impact of *cycA* overexpression on the presence of byproducts, we chose to examine l-isoleucine production, because several CycA substrates (e.g., BCAAs, Ala) often appear in the culture broth after fermentation of Ile-producing strains. This occurs due to the participation of common enzymes in the biosynthetic pathways for BCAAs and similar metabolites (Figure 11). From a practical point of view, a decrease in the level of impurities in Ile production is of particular interest because these compounds are structurally similar to Ile and, therefore, complicate separation of the main product at the purification step.

An isoleucine-producing strain (M13) was constructed on the basis of the 44-3-15 Scr Ile-producing strain [42] via introduction of the *kan*-P_L_-*cycA* construct. To perform model test-tube cultivation with a significant number of byproducts, we carried out fermentation using the parental (44-3-15 Scr) and newly obtained strain (M13) in the presence of 20 µM chlorsulfuron. This chemical prevents branched-chain amino acid biosynthesis by inhibiting acetolactate synthase (a key enzyme). In this case, Ile production is reduced, whereas the production of side metabolites at the upstream stages of the metabolic pathway is enhanced. Using this model, we were able to detect the main impurities in the culture broth through TLC analysis, with significant decreases observed in the accumulation of Nva, Ala, and AABA in the strain with enhanced CycA synthesis (M13). These data are summarized in Table 2.

Val accumulation was estimated in the presence of 4 g/L threonine. Threonine addition allows reducing the Val/Ile ratio in the culture broth and detecting the impact of CycA-mediated Val import. The results are shown in Table 3.

Therefore, we can conclude that there is reduced accumulation of impurities from the isoleucine-producing strain overexpressing *cycA*, which confirms the function of CycA in our application. Moreover, the optimization of *cycA* gene expression maybe useful for different processes with a need to reduce byproducts such as Ala, BCAAs, and other CycA substrates.

## 4. Discussion

The microbial production of amino acids is a vast area where various metabolic engineering strategies have been successfully applied. The latest tendency in production is to use a combination of amino acid intracellular synthesis control and the alteration of transport systems, where the enhancement of excretion and blocking import of the target AA are commonly considered [10]. Thus, the characterization of transporters of amino acids and other important metabolites is currently an active field of research. In this work, we explored the substrate specificity, regulation of synthesis, and practical application of the CycA transporter. CycA is an inner-membrane protein that mediates the uptake of d-serine, d/l-alanine, β-alanine, glycine, and d-cycloserine. The mechanism of regulation of CycA synthesis is known to be translational repression mediated by small RNA GvcB [20].

Here, we identified branched-chain amino acids and several other compounds as substrates for CycA on the basis of the following evidence: (i) the overexpression of *cycA* increased the susceptibility of cells to toxic metabolites or synthetic toxic analogues of nontoxic metabolites; (ii) CycA oversynthesis resulted in a higher rate of import of branched-chain amino acids. Amino acid uptake was assayed using wild-type *E. coli* strain MG1655 and its derivative with enhanced CycA synthesis. We can see from the graphs that the rate of alanine and AABA import was significantly higher in the strain with overexpressed *cycA*, whereas for the set of BCAAs, the increase in uptake was not so pronounced. *E. coli* transport systems capable of BCAA import have been known for a long time and are well characterized, such as the ATP-dependent LivKHMGF system [59] and the Na^+^-dependent BrnQ system [60]. These systems were intact in our strains, and a slight difference in the uptake rate upon CycA oversynthesis can be explained by the presence of these functional systems. In contrast, CycA seems to be a main importer of alanine and probably AABA, which are structurally similar to β-Ala. This conclusion is consistent with our data, following numerous attempts to obtain β-Ala-resistant *E. coli* mutants, which were always *cycA*-deficient (data not shown). Other investigations also reported on a key role for CycA in β-Ala transport [30].

Furthermore, we assessed whether *cycA* gene expression was affected by increased intracellular levels of its substrates. The importers of amino acids and other metabolites play pivotal roles in bacterial physiology, including the maintenance of a balanced intracellular pool. This is the reason for the strict regulation of expression of corresponding genes. Generally, such regulation is mediated by transcription regulators. In some cases, it depends on the intracellular concentration of substrate amino acids. Usually, only the most essential metabolites can influence the expression level of the genes encoding the corresponding transporters. For transporters with a broad range of substrates, only a few affect synthesis at the protein level. For CycA, such regulation at the level of translation is known and is facilitated by the small regulatory RNA GcvB and chaperone protein Hfq [32]. In this case, there is negative feedback dependent upon the substrate concentration (glycine). GcvB also provides tight regulation of other genes involved in the transport of amino acids and of peptides under conditions where these molecules are in excess. To test the dependence of *cycA* expression level upon the elevation of its substrate concentration, we used a translational fusion of the *lacZ* gene with the *cycA* regulatory region that included a GcvB-complementary site. As expected, the addition of Gly to the culture medium led to a decrease in β-galactosidase activity; this indicates that our model system provides relevant results. Apparently, the concentration of other substrates (e.g., BCAA) of the transporter CycA may also influence the level of synthesis of this protein. However, the mechanism of such a regulation has not yet been described in the literature.

There is little information about the transcription regulation of *cycA* gene expression in the literature. Some ChIP-seq investigations report that two dual regulators, Lrp and ArcA, inhibit *cycA* transcription, and that the Nac dual regulator activates transcription [33,36,37]. As for Nac, there are microarray data indicating that this regulatory protein activates the transcription of the *cycA* gene as well as the transcription of the upstream gene *fklB* encoding peptidyl-prolyl-cis-trans isomerase [61]. In turn, Nac transcription is controlled by NtrC, a nitrogen regulatory protein C [61]. ArcA is a response regulator and is activated as a DNA-binding protein by phosphorylation. This protein forms with the membrane-associated sensor kinase ArcB, a two-component signal transduction system ArcAB, which is involved in the response to changes in respiratory growth conditions and plays an important role in anaerobic repression of genes related to aerobic metabolism (for review see [62]). Thus, studying the effects of these regulatory factors on *cycA* gene transcription requires a complex in-depth analysis of each of these regulatory cascades separately and in combination.

In this work, we focused on studying the effects of Lrp and Crp on *cycA* gene transcription and other possible ways of regulating *cycA* gene expression. A study of *cycA* gene transcription regulation using bioinformatic tools, such as the DNA–protein interaction database DPInteract [54] and bioinformatic software “Sequence Editor and Tools” [55], suggested a list of putative regulatory proteins affecting *cycA* gene expression. We examined the set, including RpoS, CRP, LRP, IHF, HNS, and FarR. As a first step, to verify the results of in silico analysis, we tested whether the individual deletions of genes encoding these factors affected transcription from the P*_cycA_* promoter. At this stage, RpoS, Crp, IHF, and HNS appeared as putative activators, whereas Lrp repressed *cycA* transcription in the absence of Leu. Furthermore, we studied the ability of the proteins from the crude extract of *E. coli* cells to bind to the region upstream of the *cycA* gene. Among the three detected DNA–protein complexes, we detected the presence of the Crp factor at the highest molecular weight band, and the RpoS and Lrp regulators at the middle and lowest molecular weight bands, respectively (Figure 9).

LRP (leucine-responsive regulator protein) is a global dual transcription regulator. Its regulon involves at least 10% of *E. coli* genes, including those responsible for amino acid biosynthesis and catabolism, nutrient transport, and other cellular functions, such as carbon metabolism. The regulatory action of Lrp on target genes is often modulated by the binding of the small effector molecule leucine. As a dual regulator, Lrp is able to affect transcriptional regulation in all possible ways: by activating or inhibiting transcription whereby the addition of leucine reverses the effect; by activating or inhibiting transcription upon leucine binding; by activating or inhibiting transcription independently of the leucine-binding status. Furthermore, it was shown that Lrp often binds to promoters to enable combinatorial interactions with other regulators [63]. The gene *cycA* was identified as a member of the Lrp regulon on the basis of ChIP-chip and RNA-seq data [33,63]. In our work, we demonstrated the effect of *lrp* deletion in the presence or absence of Leu on the reporter gene expression rate from the P*_cycA_* promoter, which is evidence of the participation of this transcription factor in the regulation of CycA synthesis. Leu is the most common coregulator for Lrp; in the case of *cycA*, it removes the repressive effect of Lrp. Other amino acids are also known to affect Lrp behavior [53], e.g., Ala and BCAA, which are also among CycA substrates. Moreover, the addition of such amino acids enhanced the expression from P*_cycA_* in our experiments. Taking our results together with the literature data, we can conclude that Lrp takes part in *cycA* regulation. Thus, *cycA* can be attributed to the group of Lrp-dependent genes, which are activated by the Lrp factor in the presence of leucine in the medium. This group includes genes encoding branched-chain amino acid carrier proteins, such as the genes of the *livKHMGF* operon [64]. In this case, we can observe the existence of a positive feedback loop in the expression of the *cycA* gene, in which the substrate is transferred into a cell, thus stimulating its intracellular accumulation by increasing the level of importer. Another example of such control is the dependence of *cycA* gene expression on the concentration of the glycine (CycA substrate) in the cell provided via sRNA GcvB, which is a negative feedback loop. The *gcvB* gene expression depends on the activity of the repression factor GcvA. In the presence of Gly, GcvA is inactive, and GcvB synthesis is initiated, which leads to *cycA* translational inhibition [32,65]. This assumption fits well with the existing hypothesis that the expression of transporter genes is controlled by their own substrates.

Another putative *cycA* regulator is the cAMP receptor protein CRP (cAMP receptor protein). It is a global transcription factor that controls the activity of more than 180 genes in *E. coli* cells [34,35]. Unlike the Lrp factor, the main regulator of the metabolism of amino acids and nitrogen-containing compounds, the Crp protein mainly regulates the catabolism of carbon sources [64]. The Crp can activate or repress the transcription of certain genes, and its mode of action depends on the mode of interaction of Crp with the target gene regulatory region [58,66]. The Crp-dependent promoters are divided into three classes on the basis of their mode of interaction with the transcription factor [66]. Class I promoters contain a Crp-binding site located upstream of the DNA site for RNA polymerase, and transcription is activated following Crp binding. Class II promoters contain a Crp-binding site overlapping with the “−35” region, and transcription may be activated, as well as repressed, after Crp binding. Class III promoters contain two DNA sites for Crp with different distances between them and different locations relative to the DNA site for RNA polymerase. Furthermore, direct and indirect activation mechanisms can be realized because some transcription factors belonging to the Crp regulon can control the expression of a gene of interest by directly binding with its regulatory region.

According to the in silico prediction, the Crp binding site is located at positions −93 to −114 upstream of the transcription start site; thus, the promoter of the *cycA* gene may belong to the group of Class I Crp-dependent promoters. Accordingly, we would expect activation of transcription by the Crp factor. As mentioned, deletion of the *crp* gene resulted in a decrease in reporter gene expression controlled by the *cycA* regulatory region; therefore, Crp can be proposed as an activator of *cycA* expression, and such an effect can be direct or indirect. Indirect Crp action can be realized through the Hfq protein that works in complex with the GcvB small regulatory RNA at the translational level. The *hfq* gene undergoes downregulation by the Crp protein; hence, decreasing the Hfq amount may lead to derepression of *cycA* translation. We tested this possibility using transcriptional fusion. The fusion was arranged in such a way that the *lacZ* coding frame was located immediately after the P*_cycA_* promoter. Therefore, there was no *cycA*-coding sequence fragment in the fusion construct to exclude the possibility of GcvB influence. As the influence of *crp* deletion on the expression level was retained, we rejected the possibility of indirect regulation via Hfq. Furthermore, we investigated more precisely the binding of Crp to the region upstream of the *cycA* gene promoter. By means of EMSA experiments, we obtained evidence for the interaction of the pure Crp factor with the *cycA* regulatory region, particularly with the predicted DNA fragment. In conclusion, we can state that Crp directly activates *cycA* transcription and that P*_cycA_* is a Class I Crp-dependent promoter.

The results obtained are consistent with the data presented in the literature. Thus, according to microarray analysis, Crp activates the expression of *cycA* in the exponential phase when the culture grows on minimal medium [34]. Therefore, the results obtained in this work confirm the fact that the transcription factor Crp, a key regulator of carbon uptake and catabolism, also regulates the expression of many alternative substrate uptake systems and genes involved in amino acid degradation in *E. coli* [35,67].

From the view of practical application, breeding of industrially important strains producing amino acids usually includes modification(s) of cell transport systems. These modifications are concerned, as a rule, with the enhancement of efflux pumps excreting a target product. Increasing the export of a main product is a necessary step for successful strain development. However, the requirements of the downstream process in the microbial production of amino acids often necessitate decreasing the impurities in the culture broth, especially in the case of a structural similarity of these impurities with the main product. Thus, the search for approaches to lower the level of byproduct is rather important; in this context, the reuptake of such compounds is of particular interest from the viewpoint of the entire production process. Optimization of the synthesis of these proteins may solve problems related to the presence of undesirable byproducts in the culture broth and simplify further purification of the product of interest.

A demonstration of the benefits of CycA synthesis optimization was performed using the example of the isoleucine production process. The distinctive feature of the biosynthetic pathway of BCAAs is the exploitation of common enzymes with double specificity for the synthesis of both valine and isoleucine. Thus, for example, the key isozymes AHASes (acetohydroxyacid synthases) catalyze the biosynthesis of α-aceto-α-hydroxybutyrate, for the isoleucine pathway from pyruvate and α-ketobutyrate, and of α-acetolactate, for the valine pathway from two molecules of pyruvate. Each of the subsequent steps is catalyzed by an enzyme common to the precursors of Ile or Val. Therefore, enhancement of their activity leads to the accumulation of both the main product and the byproducts (Figure 11). This is the reason for the high amount of BCAA byproducts, including a product of Ile precursor amination, AABA, observed in the Ile production process. Such impurities are structurally very similar to Ile. This circumstance complicates obtaining the main product with a desirable degree of purity. Decreasing the amount of byproducts during the fermentation process may simplify purification of the culture broth and reduce its cost. The standard approaches to decreasing impurities, such as altering the substrate specificity of corresponding enzymes or changing the pools of central metabolites used in this pathway, are not useful in the case of structurally similar compounds produced by common reactions. The proposed alternative approach is to enhance the import of impurities back into cells. CycA, being an importer for the range of BCAAs and similar compounds, is capable of reuptake of undesirable impurities in the Ile production process. As shown here, the overexpression of *cycA* in an isoleucine-producing strain resulted in a significantly reduced accumulation of BCAA byproducts in the model fermentation process. The function of CycA as an importer of BCAAs and similar metabolites was confirmed. From the view of practical application, we showed the usefulness of transport optimization, particularly influx optimization, to obtain an appropriate culture broth composition.

## 5. Conclusions

As a result of our investigation, the range of known substrates for CycA permease was expanded; thus, it was shown that CycA mediates the uptake of Val, Nva, AABA, Leu, Ile, and 2KB. The influence of substrate concentration on CycA synthesis was demonstrated. Evidence for the participation of global transcription factors, including Lrp and Crp, in the regulation of *cycA* gene expression level, was obtained. Moreover, oversynthesis of the CycA permease was applied during the microbial production of Ile and enabled a significant reduction in the number of undesirable impurities. Such an effect is important for the whole process of Ile production, including the purification of the target substance. The described strategy for reducing byproduct accumulation during microbial production based on enhancing the reuptake of corresponding compounds has the potential for successful application in production of not only Ile, but also a wide range of cellular metabolites.

## 6. Patents

Hook, Ch.D.; Samsonov, V.V.; Eremina, N.S.; Stoynova, N.V.; Sidorova, T.S. Method for producing l-isoleucine using a bacterium of the family Enterobacteriaceae having overexpressed the *cycA* gene. Patents US 9,896,704; EP 3 085 705; JP 6766426.

## Figures and Tables

**Figure 1 microorganisms-10-00647-f001:**
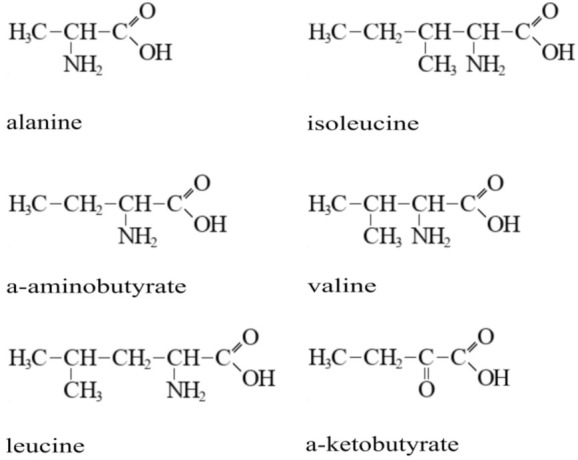
Structural formulas of l-alanine and structurally analogous compounds.

**Figure 2 microorganisms-10-00647-f002:**
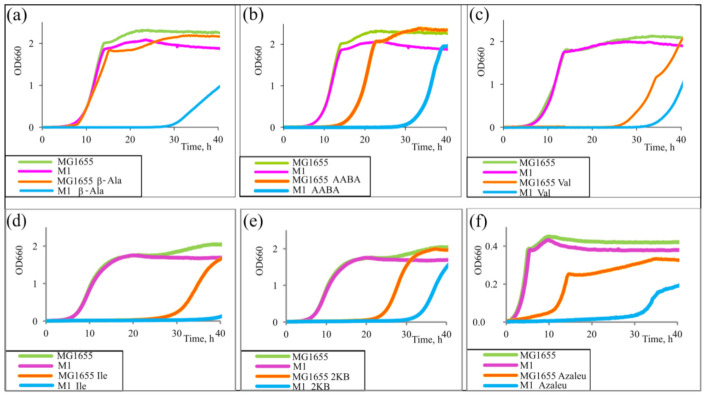
Growth inhibition assay. Strains *E. coli* K12 MG1655 and M1 (MG1655 *cat*-P_L_-*cycA*) were incubated in parallel in minimal medium (M9) containing (**a**) 50 mg/L β-alanine (β-Ala), (**b**) 50 mg/L α-aminobutyrate (AABA), (**c**) 0.5 mg/L valine (Val), (**d**) 15 g/L isoleucine (Ile), (**e**) 150 mg/L α-ketobutyrate (2KB), or (**f**) 500 mg/L azaleucine (Azaleu) at 37 °C for 40 h or without additions as a control. The data are the mean values of three replicates.

**Figure 3 microorganisms-10-00647-f003:**
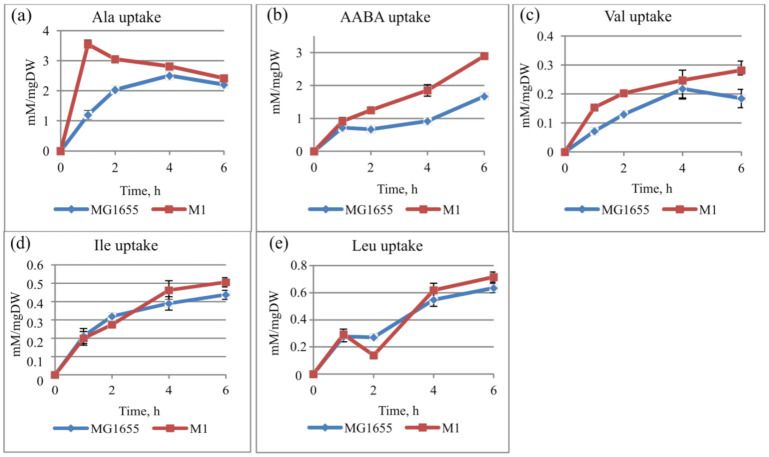
Time courses of amino acid uptake. Strains *E. coli* K12 MG1655 and M1 (MG1655 *cat*-P_L_-*cycA*) were incubated in parallel in N^−^ minimal medium containing (**a**) 50 mM alanine (Ala), (**b**) AABA, (**c**) valine, (**d**) isoleucine, or (**e**) leucine (Leu) at 37 °C for 6 h. The uptake rate (mM/mg DW) was calculated as the consumed AA (the extracellular concentration was subtracted from the initial one) normalized to the dry weight (mg DW) value. The data are the mean values of three replicates with standard deviations.

**Figure 4 microorganisms-10-00647-f004:**
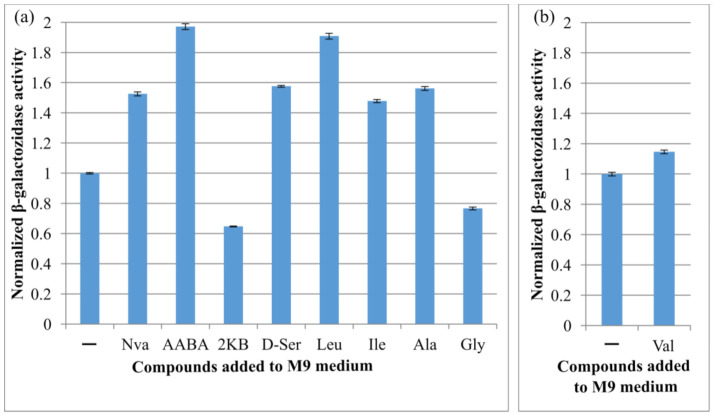
Influence of different amino acids on the expression level of *lacZ* gene under P*_cycA_* regulatory region. Strain (**a**) M2 or (**b**) M3, possessing the *cat*-P*_cycA_*-*lacZ* translational fusion, was incubated in parallel in minimal medium (M9) with or without addition of 10 mM of the indicated compound at 37 °C for 5 h, and the β-galactosidase activity was measured. The values were normalized to the activity level in M9. The data are the mean values of three replicates with standard deviations.

**Figure 5 microorganisms-10-00647-f005:**
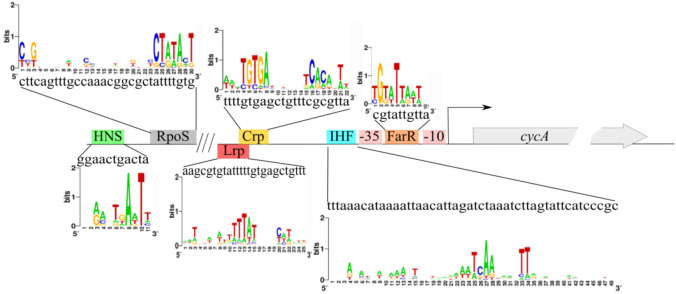
Schematic view of *cycA* regulatory region with the binding sites of putative transcription factors predicted using a bioinformatics approach. Sequence logos were created using the web-based application WebLogo (http://weblogo.berkeley.edu/, accessed on 19 October 2021).

**Figure 6 microorganisms-10-00647-f006:**
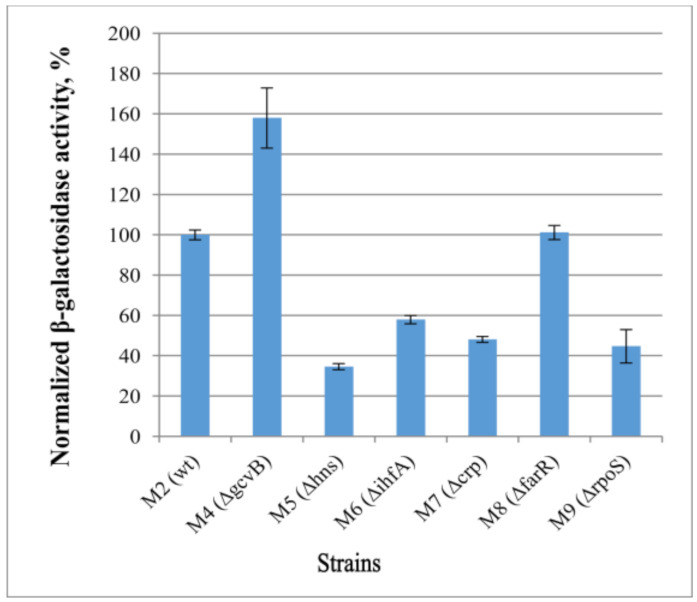
Influence of deletions of various regulatory factors on the expression level of *lacZ* gene under control of the P*_cycA_* regulatory region. The initial strain M2 (MG1655 *cat*-P*_cycA_*-*lacZ*) and its derivatives lacking various transcriptional regulators (M4–M9) were incubated in LB medium at 37 °C, and the β-galactosidase activity was measured at the exponential growth phase. The values were normalized at the activity level of the M2 strain. The data are the mean values of three independent experiments, where each assay was performed in triplicate, shown with standard deviations.

**Figure 7 microorganisms-10-00647-f007:**
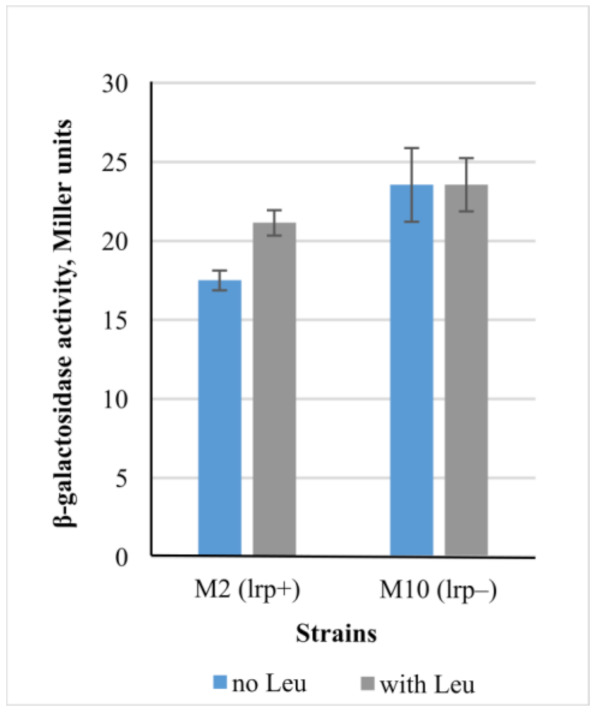
Influence of Lrp in the presence or absence of Leu on the expression level of *lacZ* gene under control of P*_cycA_* regulatory region. Strains M2 (*lrp*+) and M10 (*lrp*−) (with or without *lrp* gene, respectively) were incubated in M9 supplemented or not with 10 mM Leu at 37 °C, and the β-galactosidase activity was measured at the exponential growth phase. The data are the mean values of three independent experiments, where each assay was performed in triplicate, shown with standard deviations.

**Figure 8 microorganisms-10-00647-f008:**
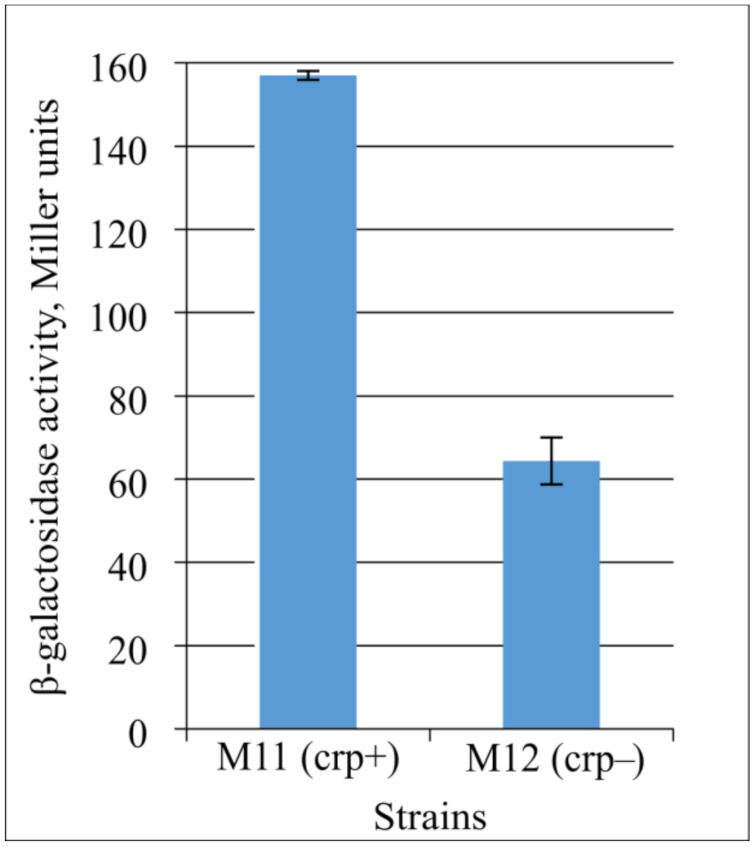
Influence of Crp on the expression level of *lacZ* gene under P*_cycA_* regulatory region without GcvB-binding site. The strains M11 (MG1655 ΔP*_lac_*::*cat*^exc^-P*_cycA_*-5′-UTR*_lacZ_*-*lacZ*, *crp*+) and its Δ*crp* derivative (M12, *crp*−) were grown in LB medium to the late exponential phase and assayed for β-galactosidase activity. The data are the mean values of three independent experiments, where each assay was performed in triplicate.

**Figure 9 microorganisms-10-00647-f009:**
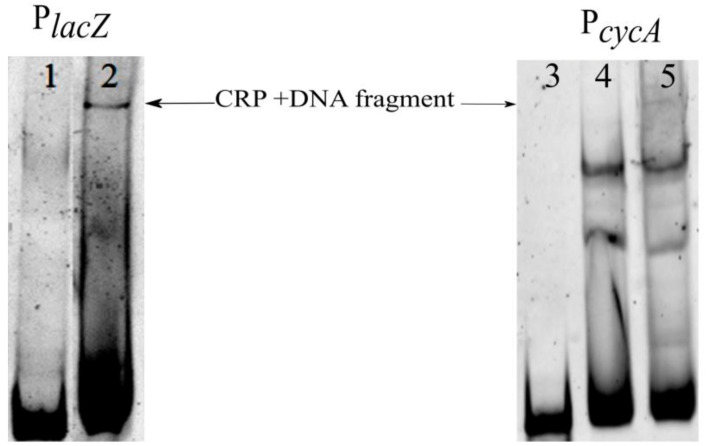
Results of electrophoretic mobility shift assay using P*_lacZ_* or P*_cycA_* DNA fragments with *E. coli* crude extract. 1, 3—free probe; 2, 5—fragment + *E. coli* K12 MG1655; 4—fragment + *E. coli* K12 MG1655 Δ*crp.*

**Figure 10 microorganisms-10-00647-f010:**
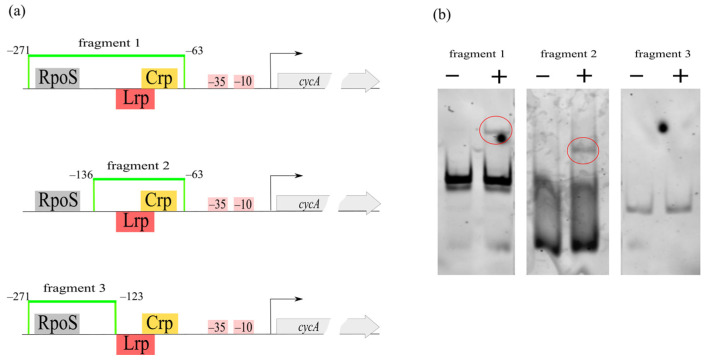
(**a**) A schematic view of *cycA* regulatory region with predicted regulator binding sites and DNA fragments used in EMSA with pure Crp protein. (**b**) EMSA results. (−) no Crp, only free probe; (+) samples with Crp; bands corresponding to retardation observed for fragments 1 and 2 are circled.

**Figure 11 microorganisms-10-00647-f011:**
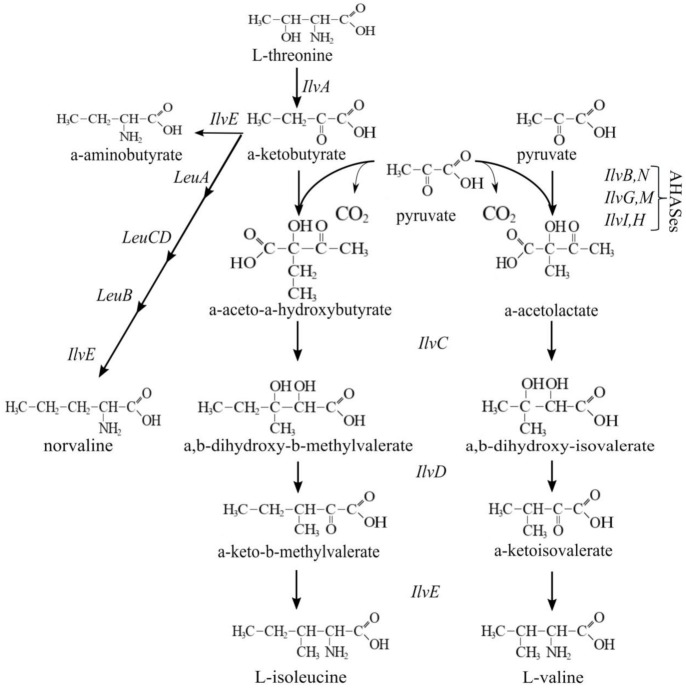
Schematic of BCAAs biosynthesis.

**Table 1 microorganisms-10-00647-t001:** Bacterial strains and plasmids used in this study.

Strain or Plasmid	Description	Reference or Source
MG1655	*Escherichia coli* K12, wild-type	VKPM ^a^ B6195
BW25113 *cat*-P_L_-*yddG*	*E. coli* K12, *cat*-P_L_-*yddG*	[39]
M1	MG1655 *cat*-P_L_-*cycA*	This work
M2	MG1655 *cat*-P*_cycA_*-*lacZ*	This work
K12 2Δ P_L_-*ilvBN*^fbr^	*E. coli* K12 ∆*ilvGM* ∆*ilvIH* P_L_-*ilvBN*^fbr^	[40]
M3	K12 2Δ P_L_-*ilvBN*^fbr^ *cat*-P*_cycA_*-*lacZ*	This work
BW25113 Δ*gcvB*	*E. coli* K12, Δ*gcvB*	KEIO collection ^b^
M4	BW25113 Δ*gcvB cat*-P*_cycA_*-*lacZ*	This work
BW25113 Δ*hns*	*E. coli* K12, Δ*hns*	KEIO collection ^b^
M5	BW25113 Δ*hns cat*-P*_cycA_*-*lacZ*	This work
BW25113 Δ*ihfA*	*E. coli* K12, Δ*ihfA*	KEIO collection ^b^
M6	BW25113 Δ*ihfA cat*-P*_cycA_*-*lacZ*	This work
BW25113 Δ*crp*	*E. coli* K12, Δ*crp*	KEIO collection ^b^
M7	BW25113 Δ*crp cat*-P*_cycA_*-*lacZ*	This work
BW25113 Δ*farR*	*E. coli* K12, Δ*farR*	KEIO collection ^b^
M8	BW25113 Δ*farR cat*-P*_cycA_*-*lacZ*	This work
BW25113 Δ*rpoS*	*E. coli* K12, Δ*rpoS*	KEIO collection ^b^
M9	BW25113 Δ*rpoS cat*-P*_cycA_*-*lacZ*	This work
BW25113 Δ*lrp*	*E. coli* K12, Δ*lrp*	KEIO collection ^b^
M10	BW25113 Δ*lrp cat*-P*_cycA_*-*lacZ*	This work
M11	MG1655 *cat*-P*_cycA_*-5′-UTR*_lacZ_*-*lacZ*	This work
M12	MG1655 Δ*crp cat*-P*_cycA_*-5′-UTR*_lacZ_*-*lacZ*	This work
SA500/pHA5	*E. coli* K12, *rpsL his su^−^* [pHA5, contains *crp* gene]	[41]
44-3-15 Scr	Ile producing strain	[42]
M13	44-3-15 Scr *cat*-P_L_-*cycA*	This work
pKD46	oriR101, repA101ts, araC, ParaB-[γ, β, exo of phage λ], ApR; used as a donor of λRed-genes to provide λRed-dependent recombination	[43]
pMW118-Cm^R^	oriR101, repA, MCS, Ap^R^, Cm^R^ λattR-*cat*-λattL; used as a donor of λXis/Int-excisable Cm^R^ marker	[44]

^a^ VKPM, The Russian National Collection of Industrial Microorganisms; ^b^ [45].

**Table 2 microorganisms-10-00647-t002:** Influence of *cycA* overexpression on the accumulation of impurities by Ile-producing strain 44-3-15 Scr.

Ile-Producing Strain	Ile, mM	Impurities, % of Ile		
		Nva	Ala	AABA
44-3-15 Scr	2.0 ± 0.2	390 ± 2	260 ± 3	362 ± 15
44-3-15 Scr *kan*-P_L_-*cycA* (M13)	0.8 ± 0.1	110 ± 3 0	<39	<12

**Table 3 microorganisms-10-00647-t003:** Influence of *cycA* overexpression on Val accumulation by Ile-producing strain 44-3-15.

Ile-Producing Strain	Ile, mM	Val, % of Ile
44-3-15 Scr	105 ± 3	26.2 ± 0.7
44-3-15 Scr *kan*-P_L_-*cycA* (M13)	107 ± 8	19 ± 3

## Data Availability

The analyzed data presented in this study are included within this article. Further data are available on reasonable request from the corresponding author.

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
