# Peer review of "The Escherichia coli Amino Acid Uptake Protein CycA: Regulation of Its Synthesis and Practical Application in l-Isoleucine Production"

_microorganisms, 2022, doi:10.3390/microorganisms10030647_

Round 1

Reviewer 1 Report

In this manuscript, authors implement a detailed functional characterization of the amino acid transporter CycA. They have convincingly justified this endeavour by stressing the importance of studying amino acid uptake for reducing DSP in industrial processes. CycA is therefore studied using several techniques, including also growth inhibition assays and transcriptional regulation. Overall, they successfully add novel knowledge on this transporter, which will be beneficial to the scientific community.

I have minor comments that authors should address in a revisioned version of the manuscript:

L 269-270: authors mention arbitrary concentrations of amino/keto acids, claiming their inhibition effect. However, no test of their inhibition (which could justify the concentration chosen) was performed. For example, the azaleu was used at 500 mg/L, but this resulted in a very poor growth, also of the control (Fig. 2f). Please clarify.

L 288: the growth-analysis of Fig. 2 involved beta-alanine, but the authors measured "Ala" (which refers to alpha-alanine) for the uptake experiment. They should repeat the experiment on beta-alanine to claim

Figure 2: yellow growth curves are very difficult to detect from the white background. Please change.

Figure 3: I am not convinced of the values plotted in the y-axis, as the OD is not always linearly correlated to the real biomass concentration. Instead, I think authors should plot the mM of the AAs on the y-axis. In case the variation is little, they might have to repeat the experiment over a longer period of time (12 h at least, as suggested by growth curves of fig. 1). Moreover, please avoid the use of bended lines while drawing graphs. Instead, change the line format to straight lines.

Figure 8: use integer numbers for the y-axis as in the other figures

Reviewer 2 Report

In the manuscript, the authors have successfully characterized and expressed the CycA in engineered E. coli. The protein CycA exhibited high sensitivity to branched-chain amino acids and their structural analogs. Moreover, the role of some transcription factors in cycA expression, including Lrp and Crp, was studied using a reporter gene construct. This work demonstrated the importance of uptake systems with respect to their application in metabolic engineering. Overall, the manuscript is technically sound and the research ideas appear justified. Nevertheless, there are many mistakes/typos and grammatical errors including sentence organization throughout the manuscript.

Listed are some comments regarding the submitted manuscript:

  1. Please check all “Error! Reference source not found” in the text. The number of tables and figures should be mentioned in a specific line. For example P. 3, line 122: Error! Reference source not found → Table 1.
  2. The constructions of the mutant strains have been already reported previously. Although the results from the study are positive, the paper looks not that novel.
  3. 1, line 31: Although E. coli strains are among the most frequently used bacterial hosts for the production of valuable compounds, including amino acids. However, Corynebacterium glutamicum is known as an industrial microbe traditionally used for the production of amino acids. Therefore, it would be better to explain the reason why E. coli strain MG1655 was chosen as the host strain in this research?
  4. 1, line 38: What’s “As mentioned”? it would be better to specifically mention.
  5. 2, lines 61-73: There are several efficient approaches to metabolic engineering of E. coli. it would be better to explain the reason why application of new specific amino acid exporters is important?
  6. 3, lines 100-105: Were there any studies to ArcA (anoxic redox control), and Nac (nitrogen assimilation control)? Why these regulatory factors are not used in this research?
  7. 3, line 123: SOB → Super Optimal Broth (SOB).
  8. 4, line 130 - P. 5, line 188: the manuscript is technically sound and the research method appears justified. However, the detailed method should be provided in Supplementary data.
  9. 4, line 141, 148, 161; P. 5, line 165, 171, 178: E. coli → E. coli
  10. 8, lines 269-270: Please provide relative reference papers about inhibiting concentrations of β-Ala, Ile, AABA, azaleucine, Val, or L 2KB?
  11. 11, line 334, figure 4: what’s the label of x-axis?
  12. 13, line 382, figure 6: what’s the label of x-axis?
  13. 13, line 390, figure 7: what’s the label of x-axis?
  14. 14, line 426, figure 8: what’s the label of x-axis?

The paper needs to be fully revised in order to highlight its purpose and it should be accepted for publication after careful revision.

Round 2

Reviewer 2 Report

I have agreed to the revised manuscript for publication.